

# Effects of perioperative intravenous esketamine on postoperative recovery quality in adult patients: a systematic review and meta-analysis

Zibang He, Jin Wu, Chun Yang and Peng Jiang

Department of Anesthesiology, Affiliated Hospital of Jiangsu University, Jiangsu University, Zhenjiang, Jiangsu, China

## ABSTRACT

**Objective.** This study aimed to evaluate the effects of perioperative intravenous esketamine on the quality of postoperative recovery in adult patients.

**Methods.** The primary outcome was post-anesthesia care unit (PACU) stay time. Secondary outcomes included extubation time, Quality of Recovery-40 (QoR-40) and Quality of Recovery-15 (QoR-15) scores, pain scores at 24 hours postoperatively, incidence of postoperative nausea and vomiting (PONV), postoperative sleep quality, anxiety and depression, and mental side effects. A comprehensive literature search was conducted in PubMed, EMBASE, Web of Science, the Cochrane Library, and China National Knowledge Infrastructure from inception through April 20, 2023, with an update on May 18, 2024. The study was registered on PROSPERO (Registration number: CRD42023399290). Mean differences (MD) or relative risks (RR) with 95% confidence intervals (CI) were used to estimate effect sizes. Meta-analysis was performed using RevMan 5.3 and Stata 16.0 software.

**Results.** Nineteen studies involving 1,967 patients were included. No significant differences were observed between the esketamine and control groups in PACU stay time (MD = 0.99, 95% confidence interval (CI) [−2.31–4.30], $P = 0.56$) or extubation time (MD = 1.30, 95% CI [−1.10–3.17], $P = 0.34$). However, the esketamine group showed significantly higher postoperative QoR-40 scores (MD = 9.40, 95% CI [6.12–12.69], $P < 0.00001$) and QoR-15 scores (MD = 7.43, 95% CI [3.97–10.88], $P < 0.0001$) compared to the control group.

**Conclusion.** Perioperative intravenous esketamine does not significantly affect PACU stay time, extubation time, or the incidence of postoperative mental side effects. However, it can reduce pain within 24 hours after surgery, improve sleep quality, decrease the incidence of PONV, and enhance postoperative recovery as reflected by higher QoR scores.

Corresponding authors
Chun Yang, chunyang@njmu.edu.cn
Peng Jiang, doctorjp@163.com

# INTRODUCTION

With the introduction and development of the Enhanced Recovery After Surgery (ERAS) concept (*Ljungqvist, Scott & Fearon, 2017*), surgeons and anesthesiologists are increasingly

focusing on the quality of postoperative recovery. Multi-modal analgesia is currently acknowledged as an effective perioperative analgesic strategy that ensures pain relief while minimizing the side effects associated with opioid use (*Azmat, 2007*). Ketamine, an N-methyl-D-aspartate (NMDA) receptor blocker, is the only intravenous anaesthetic available clinically that provides analgesia and is considered an important component of multi-modal analgesia (*Schwenk et al., 2018*). Esketamine, the dextroform of ketamine, exhibits a higher affinity for NMDA receptors compared to ketamine. It requires only half the dose of ketamine to achieve the same anaesthetic and analgesic effects, has a shorter half-life, and is associated with fewer mental side effects (*White et al., 1985*; *Weiskopf, Nau & Strichartz, 2002*). A growing body of research has focused on the perioperative adjunctive analgesic effects of esketamine (*Argiriadou et al., 2004*; *Lahtinen et al., 2004*). However, the efficacy data of esketamine in terms of quality of postoperative recovery are not centralized. Therefore, we performed a meta-analysis of randomized controlled trials to assess the influence of perioperative adjunctive analgesia with esketamine on the quality of postoperative recovery in adult patients.

## MATERIALS AND METHODS

### Search strategy and selection criteria

Initially, two independent reviewers (Zibang He and Jin Wu) carefully screened the titles and abstracts of records retrieved from the database. Articles that met the initial criteria or whose titles and abstracts were uncertain were retrieved for full-text evaluation. During the process, prior reviewer (Zibang He and Jin Wu) discrepancies are resolved through mutual discussion and, if necessary, in consultation with a third senior reviewer (Peng Jiang). This study was conducted in accordance with the PRISMA Statement (*Higgins, 2008*) and the Cochrane Handbook (*Moher et al., 2009*). It was registered on the PROSPERO website (Registration number: CRD42023399290). Computer searches were performed in databases including PubMed, Web of Science, Embase, Cochrane Library, and CNKI. The search period extended from the inception of the databases to April 20, 2023. Randomized controlled trials investigating perioperative esketamine analgesia were sought. Additionally, the references of the included studies were traced, and a search update was performed on May 18, 2024. Subject headings combined with free words were used for the search. The English search terms were "esketamine, S-ketamine, pain, analgesia, opioids, postoperative, recovery quality, QoR" while the Chinese search terms were the same.

### Inclusion and exclusion criteria

Inclusion criteria: (1) the study included adult patients undergoing general anesthesia for various surgeries. (2) The study design was a randomized controlled trial. (3) The intervention involved perioperative administration of esketamine as an adjunct to analgesia, with a placebo or other drugs used in the control group. (4) The primary outcome was post-anesthesia care unit (PACU) stay time, and secondary outcomes included extubation time, 24-hour postoperative pain scores at rest and during movement, incidence of nausea and vomiting after surgery, postoperative sleep quality, anxiety and depression scores,

mental side effects, Quality of Recovery (QoR)-40 scores and QoR-15 scores at 48 h postoperatively.

Exclusion criteria: (1) duplicate literature, reviews, abstracts, case reports, *etc*. (2) Incomplete data in the literature.

### Literature screening and quality assessment

Two researchers independently conducted a rigorous screening of the retrieved literature based on the inclusion and exclusion criteria. Duplicate literature was excluded, and obviously irrelevant literature was excluded based on the title and abstract. The final selection of included literature was determined through a thorough examination of the full text. The quality of the included literature was assessed using the Cochrane Reviewers' Handbook, which evaluated random sequence generation, allocation concealment, blinding of implementation and outcome assessment, integrity of study data, selective reporting of study results, and other sources of bias. The risk of bias was classified into three levels: "low risk," "high risk," and "unclear." These assessments were performed independently by two researchers, with any discrepancies resolved through discussion with a third party.

### Statistical analysis

Data analysis was conducted using RevMan 5.3 and STATA16.0 software. Continuous variables were presented as mean difference (MD) with a 95% confidence interval (CI). Dichotomous variables were expressed as relative risks (RR) with 95% CIs. Heterogeneity was assessed using the Q test, and quantification was performed using the $I^2$ statistic. Heterogeneity was considered significant if $I^2 \geq 50\%$, and a random-effects model was used for analysis. If $I^2 < 50\%$, no significant heterogeneity was assumed, and a fixed-effect model was used. Publication bias was evaluated using funnel plots and the Egger test. A significance level of $p < 0.05$ was considered statistically significant.

## RESULTS

### Trial selection and characteristics

After conducting a computer search and screening other resources, a total of 1,307 relevant articles were identified. After removing 1,151 duplicates and conducting a title and abstract review, 31 articles were selected. Finally, after reading the full text, 19 articles that met the criteria were included (*Brinck et al., 2021*; *Chen et al., 2022*; *Cheng et al., 2022*; *Gao et al., 2023*; *Ithnin et al., 2019*; *Li et al., 2023*; *Liu et al., 2024*; *Qiu et al., 2022*; *Wang et al., 2022a*; *Wang et al., 2023*; *Xu et al., 2023*; *Yu et al., 2022*; *Yuan et al., 2022*; *Zhang et al., 2022a*; *Zhang et al., 2022b*; *Zhang et al., 2023*; *Zhao et al., 2023*; *Zhu et al., 2022a*; *Zhu et al., 2022b*) (Fig. 1). These 19 articles encompassed a total of 1,967 patients, with 1,116 cases in the esketamine group and 851 cases in the control group. The publication period ranged from 2019 to 2024, and the sample sizes ranged from 47 to 183 cases. The included studies covered various types of surgeries, including thyroid surgery, breast surgery, thoracic surgery, abdominal surgery, and gynecological surgery. Table S1 presents the characteristics of the included studies. The risk of bias in the included studies was assessed according to the Cochrane Handbook for Systematic Reviews of Interventions. Two studies

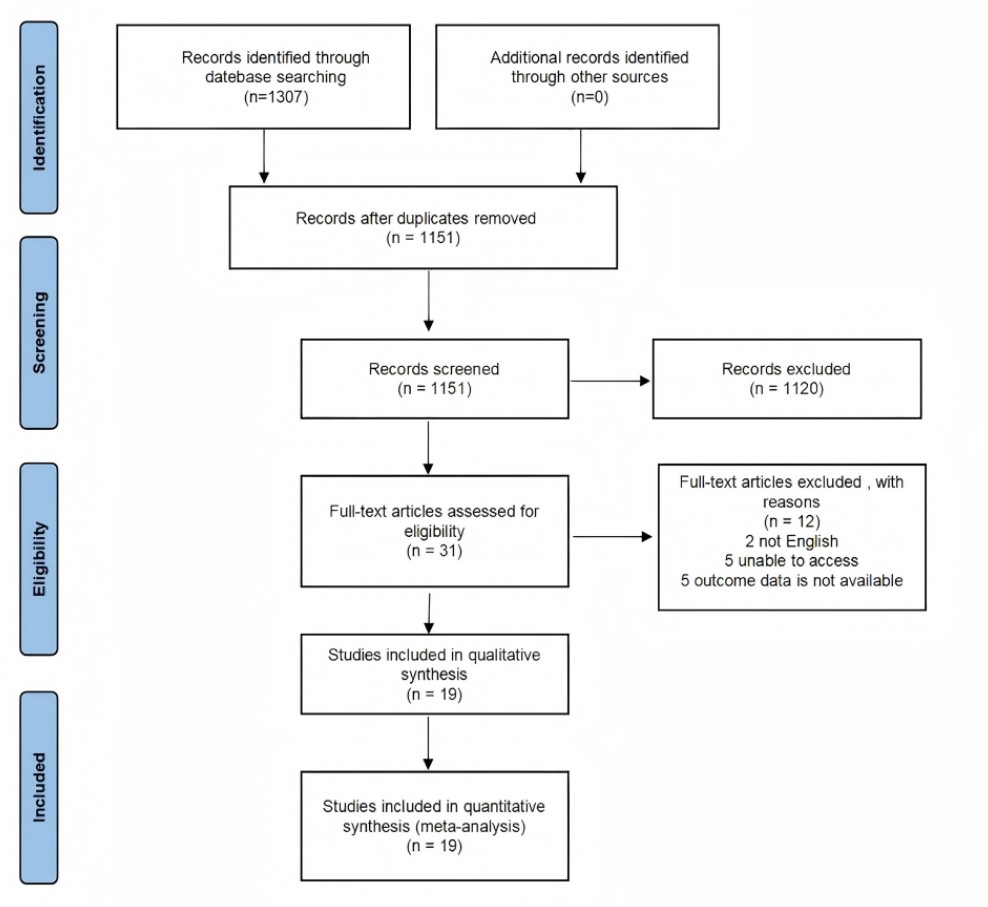

**Figure 1** **Flow chart of literature screening.** After reading the full text, 19 articles that met the criteria were included.

were classified as having a high risk of bias, 12 studies as having a low risk of bias, and the remaining studies as unclear (Fig. 2).

## PACU stay time

Ten studies involving 1,099 patients reported the post-anesthesia care unit (PACU) stay time and showed significant heterogeneity ($I^2 = 82\%$, $p < 0.00001$). Using a random-effects model, the results indicated no statistically significant difference in PACU stay time between the two groups (MD = 0.99 min, 95% CI [−2.31–4.30], $P = 0.56$). Subgroup analysis was conducted based on evidence that subanesthetic doses of esketamine did not exceed 0.35 mg/kg in a single injection or 1 mg/kg/h in continuous infusion. Regardless of the dose used, subanesthetic or non-subanesthetic, no significant difference in PACU stay time was observed between the two groups (subanesthetic dose: MD = −0.22 min, 95% CI [−4.35–4.30], $P = 0.99$; Non-subanesthetic dose: MD = 3.15 min, 95% CI [−3.84–10.14], $P = 0.38$) (Fig. 3A). Similarly, when analyzing different administration methods of esketamine, including single intravenous administration, intraoperative continuous

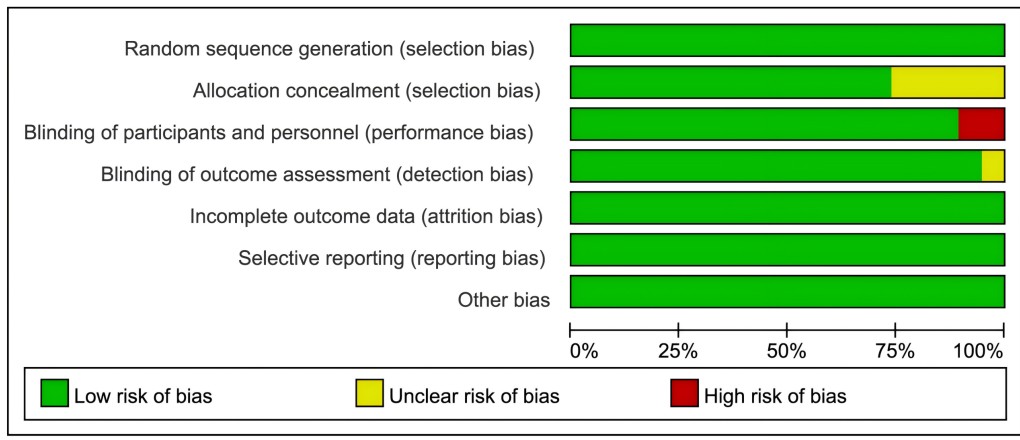

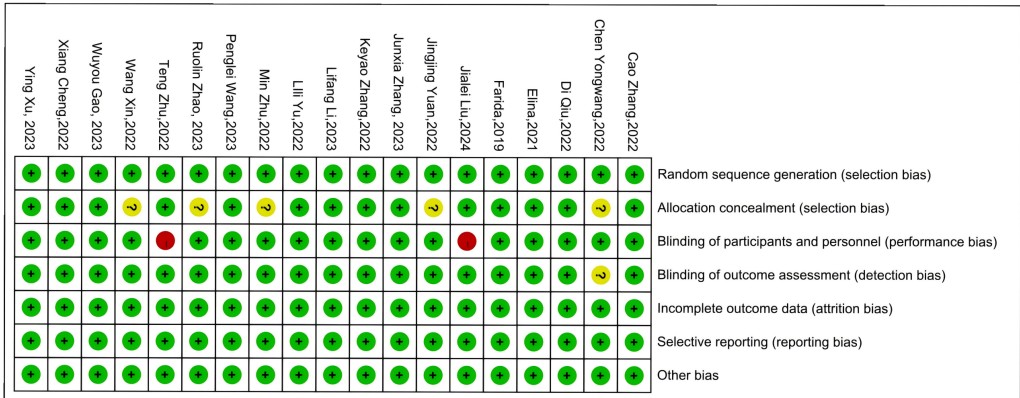

**Figure 2 Risk bias diagram for included study.** The risk of bias in the included studies was assessed according to the Cochrane Handbook for Systematic Reviews of Interventions. Two studies were classified as having a high risk of bias, 12 studies as having a low risk of bias, and the remaining studies as unclear. *Zhang et al., 2022a, Chen et al., 2022, Qiu et al., 2022, Brinck et al., 2021, Ithnin et al., 2019, Liu et al., 2024, Yuan et al., 2022, Zhang et al., 2023, Zhang et al., 2022b, Li et al., 2023, Yu et al., 2022, Zhu et al., 2022a, Wang et al., 2023, Zhao et al., 2023, Zhu et al., 2022b, Wang et al., 2022a, Gao et al., 2023, Cheng et al., 2022, Xu et al., 2023.*

infusion, or a combination of both, no statistically significant difference in postoperative PACU stay time was found between the two groups (single injection: MD = −1.65 min, 95% CI [−6.11–2.81], P = 0.47; continuous infusion: MD = −7.92 min, 95% CI [−33.25–17.41], P = 0.54; Single and continuous: MD = 2.49 min, 95% CI [−2.33–7.32], P = 0.31) (Fig. 3B).

## Extubation time

We defined extubation time as the time from when anaesthetic drugs were stopped to when the tracheal tube was removed. Postoperative extubation time was reported in eleven studies, which exhibited significant heterogeneity ($I^2 = 92\%$, $p < 0.00001$). The random-effects model was used, and the results showed no statistically significant difference in extubation time between the two groups (MD = 1.03 min, 95% CI [−1.10–3.17], P = 0.34) (Fig. 4A).

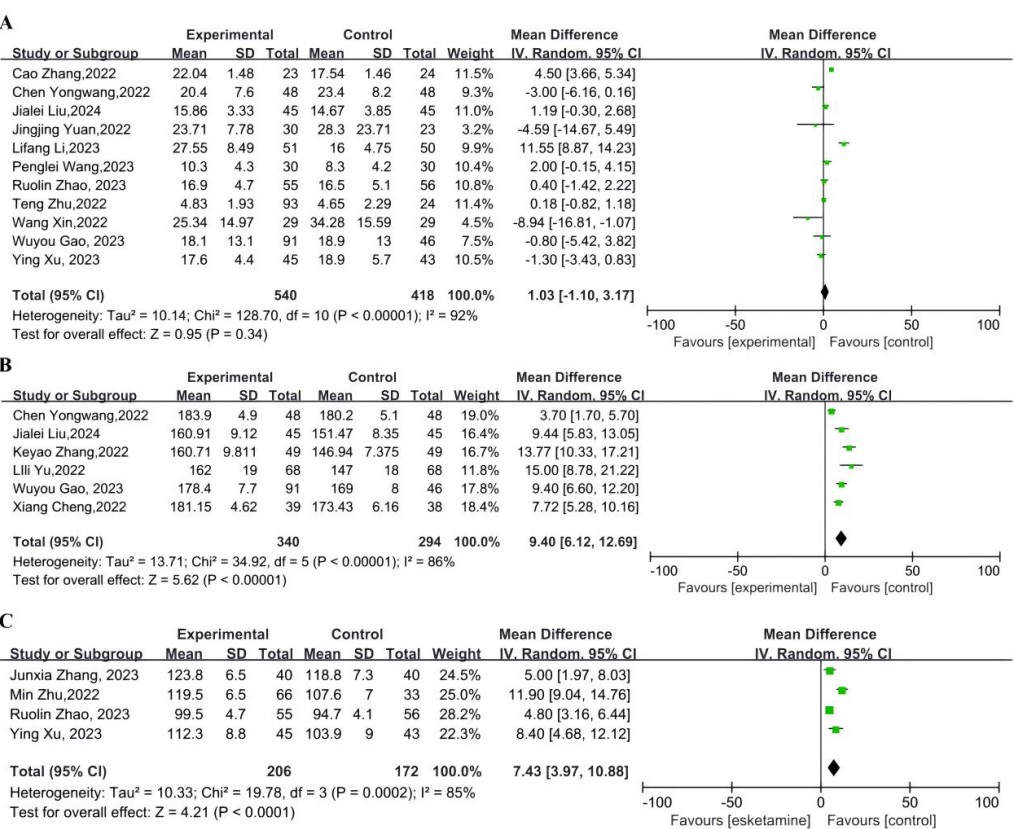

**Figure 3** **Forest plot of PACU stay time subgroup analysis (A–C).** When analyzing different administration methods of esketamine, including single intravenous administration, intraoperative continuous infusion, or a combination of both, no statistically significant difference in postoperative PACU stay time was found between the two groups. *Qiu et al., 2022, Ithnin et al., 2019, Yuan et al., 2022, Zhao et al., 2023, Zhu et al., 2022b, Wang et al., 2022a, Gao et al., 2023, Brinck et al., 2021, Zhang et al., 2022b, Yu et al., 2022.*

## QoR-40 and QoR-15 scores

Six studies reported QoR-40 scores within 48 h after surgery, demonstrating significant heterogeneity ($I^2 = 86\%$, $p < 0.00001$). Using the random-effects model, the results indicated that the QoR-40 scores in the esketamine group were significantly higher than those in the control group within 48-hour after surgery (MD = 9.40, 95% CI [6.12–12.69], $P < 0.00001$) (Fig. 4B). Four studies reported QoR-15 scores within 48 h postoperatively, demonstrating significant heterogeneity ($I^2 = 85\%$, $p = 0.0002$). Using the random-effects model, the results indicated that the QoR-15 scores in the esketamine group were significantly higher than those in the control group within 48-hour after surgery (MD = 7.43, 95% CI [3.97–10.88], $P < 0.0001$) (Fig. 4C).

## Pain scores 24 h after surgery

Eight studies reported resting pain scores at 24 h post-surgery and seven studies reported 24 h pain scores at movement postoperatively, showing heterogeneity (resting: $I^2 = 71\%$, $p = 0.001$; exercise: $I^2 = 86\%$, $p < 0.00001$). A random-effects model was utilized, and the resting and exercise pain scores at the 24-hour after surgery in the esketamine group were

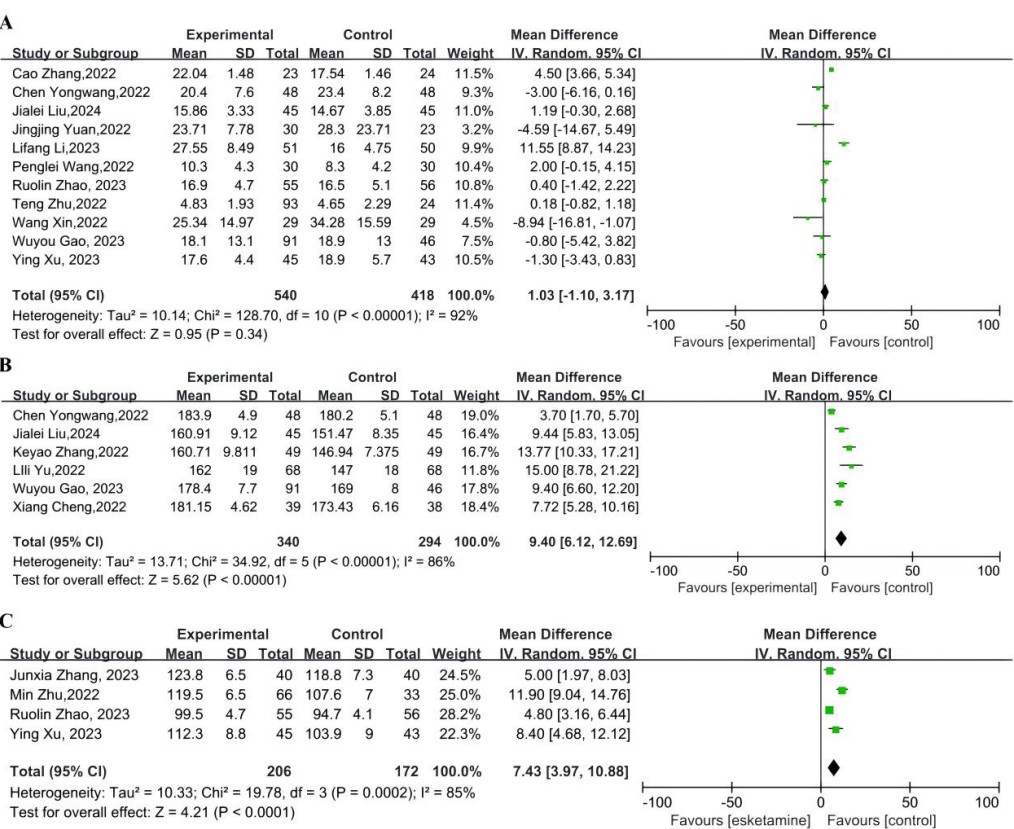

**Figure 4** **(A) Forest plot of extubation time (B) Forest plot of QoR-40 scores (C) Forest plot of QoR-15 scores.** We defined extubation time as the time from when anaesthetic drugs were stopped to when the tracheal tube was removed. Postoperative extubation time was reported in eleven studies, which exhibited significant heterogeneity. *Zhang et al., 2022a, Chen et al., 2022, Liu et al., 2024, Yuan et al., 2022, Li et al., 2023, Wang et al., 2023, Zhao et al., 2023, Zhu et al., 2022b, Wang et al., 2022a, Gao et al., 2023, Xu et al., 2023, Chen et al., 2022, Liu et al., 2024, Zhang et al., 2022b, Yu et al., 2022, Cheng et al., 2022, Zhang et al., 2023, Zhu et al., 2022a.*

lower than those in the control group (resting: MD = −0.35, 95% CI [−0.54 to −0.16], $P = 0.0004$; exercise: MD = −0.41, 95% CI [−0.70 to −0.12], $P = 0.006$) (Fig. S6).

## Postoperative anxiety and depression

Postoperative anxiety and depression scores were reported in three studies, displaying high heterogeneity (anxiety: $I^2 = 75\%$, $p = 0.02$; depression: $I^2 = 85\%$, $p = 0.001$). The Hospital Anxiety and Depression Scale (HADS) was used to evaluate patient's anxiety and depression scores in the three studies. The random-effects model was employed, and the results indicated no significant difference in postoperative anxiety and depression scores between the two groups (anxiety: MD = −0.22, 95% CI [−1.19–0.75], $P = 0.66$; depression: MD = −0.89, 95% CI [−2.21–0.43], $P = 0.19$) (Fig. S7).

## Incidence of postoperative nausea and vomiting

The incidence of postoperative nausea and vomiting was reported in twelve studies, exhibiting low heterogeneity ($I^2 = 33\%$, $p = 0.13$). Using the fixed-effect model, the

results revealed that the incidence of postoperative nausea and vomiting was 21.3% in the esketamine group and 27.9% in the conventional opioid group. The incidence of postoperative nausea and vomiting in the esketamine group was significantly lower than that in the two groups (RR = 0.80, 95% CI [0.66–0.96], $P = 0.02$) (Fig. S8A).

## Postoperative sleep quality

Three studies reported the incidence of insomnia 48 h after surgery, showing high heterogeneity ($I^2 = 91\%$, $p < 0.0001$). Employing a random-effect model, the results demonstrated lower postoperative insomnia rates in the esketamine group (RR = 0.38, 95% CI [0.17–0.87], $P = 0.02$) (Fig. S8B).

## Mental side effects

We define psychiatric side effects to include nightmares, dreaminess, hallucinations, delirium, and dissociative symptom. There were twelve reports of postoperative mental side effects in patients, exhibiting no significant heterogeneity ($I^2 = 20\%$, $p = 0.24$). The fixed-effect model was used, and the results indicated no significant difference in the incidence of postoperative mental side effects between the two groups (RR = 0.93, 95% CI [0.65–1.34], $P = 0.71$) (Fig. S8C).

## Quality of evidence

We utilized Grading of Recommendations Assessment, Development and Evaluation (GRADEpro) system to assess the quality of evidence, which evaluated risk of bias, inconsistency, indirectness, imprecision, and publication bias; The evidence quality was classified as very low, low, moderate, or high. Please refer to Table S2 for the evidence levels of primary and secondary outcomes.

## Sensitivity analysis and publication bias

A sensitivity analysis was conducted on the primary outcome PACU stay time (Fig. 5A), and the results remained robust even after excluding any study. A funnel plot was generated for PACU stay time (Fig. 5B), and the Egger's test was employed to assess publication bias ($P = 0.507$). However, since the number of studies included in our primary outcome is just 10, the presence of publication bias cannot be ruled out.

# DISCUSSION

The findings of this study demonstrated that perioperative administration of esketamine reduced postoperative pain scores at 24 h, improved sleep quality, decreased the incidence of postoperative nausea and vomiting, and enhanced the QoR scores. It did not have an impact on PACU stay time, postoperative extubation time, anxiety and depression scores, or mental side effects.

With the introduction and advancement of the Enhanced Recovery After Surgery (ERAS) concept, there is increasing focus on ensuring "patient-centered" perioperative safety, comfort, and overall quality of recovery. Anesthesiologists are particularly concerned about the quality of patients' recovery from anesthesia. The time taken for extubation

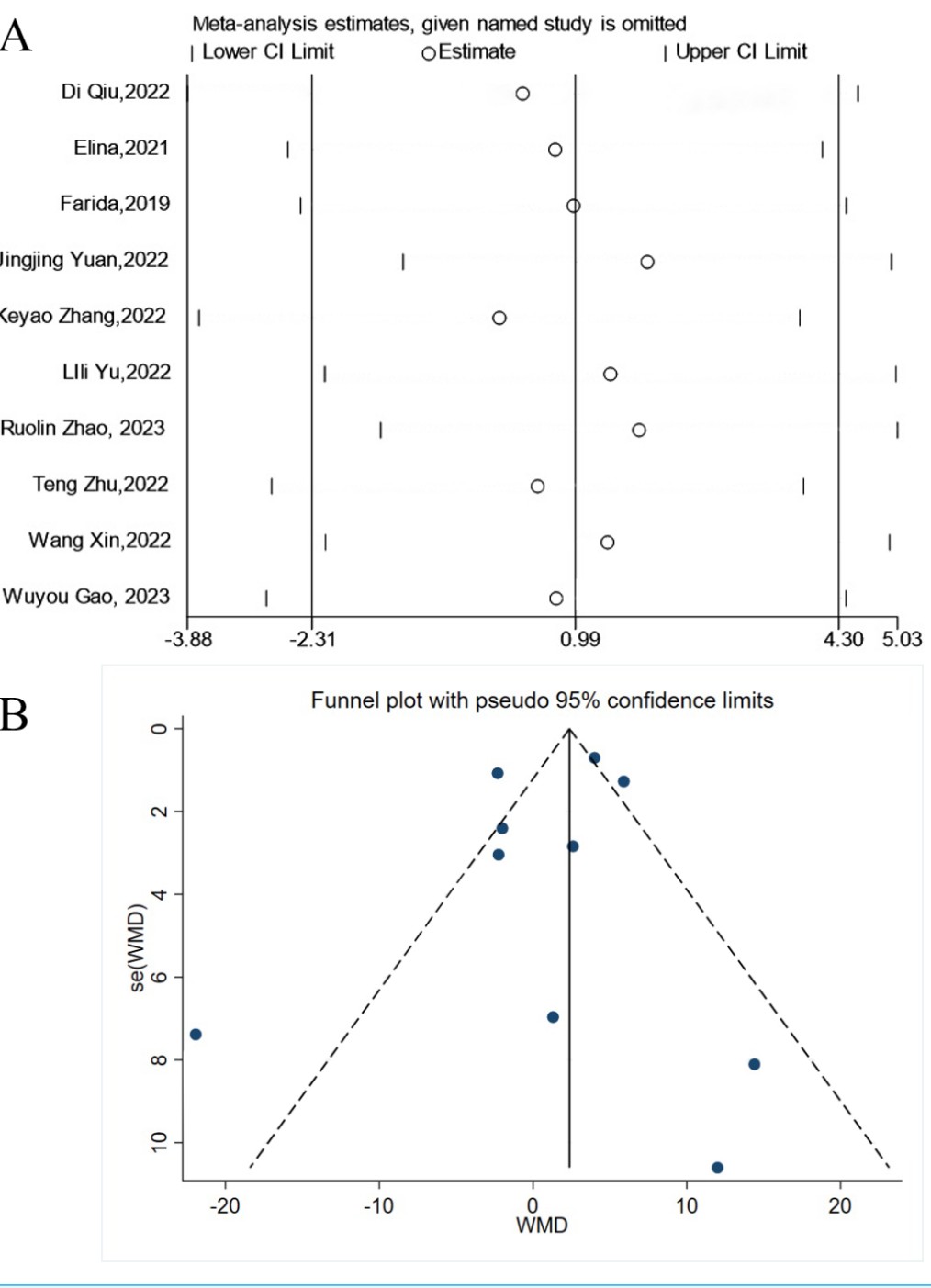

**Figure 5  Sensitivity analysis and publication bias.** A sensitivity analysis was conducted on the primary outcome PACU stay time (Fig. 5A), and the results remained robust even after excluding any study. A funnel plot was generated for PACU stay time (Fig. 5B), and the Egger's test was employed to assess publication bias (*P* = 0.507). *Qiu et al., 2022, Brinck et al., 2021, Ithnin et al., 2019, Yuan et al., 2022, Zhang et al., 2022b, Yu et al., 2022, Zhao et al., 2023, Zhu et al., 2022b, Wang et al., 2022a, Gao et al., 2023*.

after surgery and post-anesthesia care unit (PACU) stay can serve as indicators of early postoperative recovery quality for patients.

Previous research has indicated that a low dose of esketamine (0.3 mg/kg) does not affect the recovery time of children after tonsillectomy and adenoidectomy and is effective in reducing the incidence of agitation during the anesthesia recovery period (*Wang et al., 2022b*). This aligns with our findings. Our meta-analysis revealed that perioperative use of esketamine did not impact extubation time or PACU stay time following anesthesia. This could be attributed to the majority of studies included in our analysis employing subanesthetic doses of esketamine, which had minimal influence on wakefulness from anesthesia. Subanesthetic doses of esketamine were defined as a single injection not exceeding 0.35 mg/kg or a continuous infusion not exceeding one mg/kg/h (*Schwenk et al., 2018*). In our subgroup analysis, we observed that PACU stay time prolonged in the higher dose group compared to the subanesthetic dose group. One study (*Zhang et al., 2022a*) reported that perioperative intravenous esketamine prolonged extubation time, increased the risk of postoperative cognitive dysfunction, and led to an extended hospital stay in elderly patients. We believe that the prolonged recovery time from anesthesia may be associated with the sedative effect of esketamine, which is dose-dependent. Additionally, it could be attributed to the choice of continuous intraoperative infusion of ultra-short-acting remifentanil for analgesia in the control group of the study. Remifentanil's rapid elimination and short half-life resulted in control patients reaching the PACU exit criteria faster, while the half-life of esketamine is 2–3 h. These contrasting findings may be closely linked to the dose, infusion duration of esketamine, and the specific surgical population being studied.

A recent study highlighted the significance of patient-centered outcomes when evaluating the effectiveness of anesthetic interventions (*McIlroy et al., 2018*). Another study (*Lee et al., 2018*) affirmed the comprehensive nature of the QoR-40 questionnaire in measuring the quality of recovery following surgery and anesthesia from a patient-centered perspective. The QoR-40 encompasses five health dimensions: physical comfort, emotional state, physical independence, psychological support, and pain (*Léger et al., 2021*). The QoR-40 is a global measure of QoR. Besides, QoR-15 (*Stark, Myles & Burke, 2013*) is a brief scale of measuring the quality of a patient's postoperative recovery . Compared with QoR-40, it provides an equally extensive and efficient evaluation of a patient's QoR after anesthesia and surgery. A prior publication *Bowyer & Royse (2018)* established the minimum clinically important difference in QoR-40 scores as 6.3 points, indicating that an intervention during the perioperative period that could result in such a change would have a meaningful impact on improving the patient's health status. However, the study conducted by *Moro et al. (2017)* demonstrated that a low dose of intraoperative ketamine injection did not enhance the quality of postoperative recovery in patients undergoing laparoscopic cholecystectomy. In contrast to our findings, this discrepancy may be attributed to differences in the dosage and administration method of ketamine. Furthermore, it could be due to the advantage of esketamine over ketamine in perioperative adjuvant anesthesia, as esketamine exhibits twice the sedative and analgesic effects with fewer adverse effects. In comparison to the control group, patients in the esketamine group experienced reduced pain scores and improved

sleep quality 24 h postoperatively. We speculate that these factors may contribute to the higher postoperative QoR-40 and QoR-15 scores observed in the esketamine group. Sleep quality has been identified as an independent factor influencing the quality of postoperative recovery (*Niu et al., 2021*). It has been proposed that esketamine might independently contribute to improving sleep quality, in addition to its effects on postoperative pain and mood enhancement (*Li et al., 2023*).

Owing to the side effects associated with opioids, there is a growing trend towards utilizing non-opioid drugs for perioperative adjuvant analgesia. The use of a low-opioid multimodal analgesic strategy is progressively becoming an essential approach to perioperative pain management. This strategy ensures effective analgesia while minimizing the adverse effects associated with opioids, such as nausea, vomiting, respiratory depression, and opioid-induced hyperalgesia. Esketamine, an N-methyl-D-aspartate receptor antagonist, has been shown in studies to be linked with acute pain modulation and the prevention of chronic pain. The perioperative use of esketamine provides a non-opioid mechanistic pathway to analgesia. A meta-analysis involving 12 randomized controlled trials (*Wang et al., 2021*) indicated that subanesthetic intravenous administration of esketamine as an adjunct to general anesthesia effectively aids in analgesia, reducing pain intensity and opioid consumption in the immediate postoperative period, which is consistent with our study's findings. The meta-analyses have also demonstrated that perioperative intravenous administration of esketamine does not increase the risk of postoperative nausea and vomiting or psychogenic adverse events. Similarly, *Massoth et al. (2021)* revealed that a non-opioid anesthesia strategy, including the use of esketamine, does not reduce the incidence of postoperative nausea and vomiting. This could be attributed to the higher incidence of nausea and vomiting after gynecological surgery than in the general surgical population, and the application of perioperative multimodal antiemetic strategy. However, another study (*Ma et al., 2023*) reported that intraoperative addition of esketamine can improve postoperative gastrointestinal function and reduce the incidence of nausea and vomiting, our result also showed that. The inconsistency in the outcomes of the two studies may be due to the different doses of esketamine and opioids during surgery.

Furthermore, the antidepressant properties of esketamine have gained widespread recognition in recent years, with its nasal spray being employed for the treatment of treatment-resistant depression. Several studies (*Shen et al., 2023*; *Liu et al., 2021*) have demonstrated that perioperative infusion of esketamine improved postpartum depression in patients and alleviated negative mood in oncology patients during the perioperative period. However, some scholars have discovered that perioperative esketamine infusion did not reduce postoperative depression scores in patients with pre-existing depression before surgery, which aligns with the findings of our study. This outcome may be attributed to the limited impact of perioperative administration of esketamine, in small doses and for a short duration, on ameliorating negative mood in patients with long-standing depressive symptoms.

Our study examined the impact of esketamine on postoperative recovery when administered as part of a multimodal analgesic regimen. To the best of our knowledge, no meta-analysis has been conducted in this specific context. However, our meta-analysis has

several limitations. Most of the results exhibited considerable heterogeneity, potentially due to variations in surgical procedures, administration methods, doses, and timing of esketamine administration. Our analysis concentrated on the early recovery period ($\leq$48 h) due to constraints in data availability. It is recommended that future research endeavors standardize Quality of Recovery assessments at later timepoints to effectively evaluate long-term effects. Moreover, confounding factors may have influenced the results. We did not perform a subgroup analysis of total perioperative esketamine use, which is closely linked to surgical duration and the occurrence of psychiatric symptoms such as prolonged sedation, nightmares, and dissociative phenomena, often displaying a dose-dependent relationship. Despite conducting comprehensive searches across multiple databases, it is still possible that some studies meeting the inclusion criteria were inadvertently overlooked. Additionally, our meta-analysis encompassed several small-scale studies, potentially leading to an overestimation of the therapeutic effect.

## CONCLUSIONS

In conclusion, perioperative intravenous administration of esketamine did not impact PACU stay time or extubation time following general anesthesia. It effectively reduced patients' pain scores within 24 h after surgery, improved postoperative sleep quality and QoR scores, and decreased the incidence of postoperative nausea and vomiting, but did not affect the occurrence of postoperative psychiatric side effects. Given the limited quality of the studies included, conducting large-scale, high-quality randomized controlled trials (RCTs) is still necessary to further validate these findings.

### Funding
The authors received no funding for this work.

### Competing Interests
The authors declare there are no competing interests.

### Author Contributions
- Zibang He performed the experiments, prepared figures and/or tables, and approved the final draft.
- Jin Wu analyzed the data, prepared figures and/or tables, and approved the final draft.
- Chun Yang conceived and designed the experiments, authored or reviewed drafts of the article, and approved the final draft.
- Peng Jiang conceived and designed the experiments, authored or reviewed drafts of the article, and approved the final draft.

### Data Availability
This is a systematic review/meta-analysis.

## Supplemental Information

Supplemental information for this article can be found online at http://dx.doi.org/10.7717/peerj.19977#supplemental-information.

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
