# Peer review of "Effects of perioperative intravenous esketamine on postoperative recovery quality in adult patients: a systematic review and meta-analysis"

_PeerJ, doi:10.7717/peerj.19977_

## Round 0.1 · original submission · Major Revisions

Reviewer 1 ·

Basic reporting

1.The primary and secondary outcomes of the study are not clearly defined, which makes the logic of the article unclear. I would like to know whether the term“postoperative recovery quality” in the title refers to the recovery quality score (QoR) or the results of multiple dimensions?
2.Once the primary outcomes are clearly defined, their description in the abstract and results section should be prioritized and distinguished from secondary outcomes.

Experimental design

1.Regarding the results on postoperative Quality of Recovery (QoR), the study only reported data within 48 hours after surgery. It is unclear whether any studies have analyzed the results at 72 hours postoperatively.
2.The authors only conducted subgroup analyses and sensitivity analysis for the results of “PACU stay time”. However, it would be more meaningful to perform subgroup analyses on the primary outcomes.

Validity of the findings

Similar articles have already been published, so the innovation is average. Fortunately, the article includes more literature, further supporting this evidence.

Additional comments

No comment

·

Basic reporting

This meta-analysis aimed to evaluate the effects of perioperative intravenous esketamine on the quality of postoperative recovery in adult patients.
1. In the Statistical Analysis section, the authors stated: “Continuous variables were presented as mean difference (MD) or standardized mean difference (SMD) with a 95% confidence interval (CI).” However, no results involving SMD were presented. Please clarify which studies, if any, used SMD.
2. The authors mentioned that “The risk of bias was classified into three levels: ‘low risk,’ ‘high risk,’ and ‘unclear.’” However, the quality of evidence was assessed using GRADEPro, and the evidence was “classifified” as very low, low, moderate, or high. These statements appear inconsistent; please clarify the distinction between the risk of bias assessment and the GRADE quality of evidence ratings.
3. It is not recommended to choose between random-effects or fixed-effects meta-analysis based solely on the I² value. You may consider presenting results from both models for comparison.
4. Please check for minor English spelling and formatting issues, such as change “I2” to “I².”

Experimental design

Study design is Ok.

Validity of the findings

Finding is Ok.

---

## Round 0.2 · accepted · Accept

Thank you for addressing the reviewer comments and providing clear indication of the improvements made to the manuscript. Your work is now suitable for publication.

·

Basic reporting

Ok,

Experimental design

Ok

Validity of the findings

Ok

Additional comments

Ok